# A Review of Childhood Acute Myeloid Leukemia: Diagnosis and Novel Treatment

**DOI:** 10.3390/ph16111614

**Published:** 2023-11-15

**Authors:** Serena Tseng, Mu-En Lee, Pei-Chin Lin

**Affiliations:** 1School of Post-Baccalaureate Medicine, College of Medicine, Kaohsiung Medical University, Kaohsiung 807, Taiwan; srtseng@gmail.com; 2School of Medicine, College of Medicine, National Yang Ming Chiao Tung University, Hsinchu 300, Taiwan; muenl.md11@nycu.edu.tw; 3Division of Pediatric Hematology and Oncology, Department of Pediatrics, Kaohsiung Medical University Hospital, Kaohsiung 807, Taiwan

**Keywords:** acute myeloid leukemia (AML), childhood, core-binding factor (CBF), *KMT2A*/11q23 rearrangement, acute promyelocytic leukemia (APL), hematopoietic stem cell transplantation (HSCT)

## Abstract

Acute myeloid leukemia (AML) is the second most common hematologic malignancy in children. The incidence of childhood AML is much lower than acute lymphoblastic leukemia (ALL), which makes childhood AML a rare disease in children. The role of genetic abnormalities in AML classification, management, and prognosis prediction is much more important than before. Disease classifications and risk group classifications, such as the WHO classification, the international consensus classification (ICC), and the European LeukemiaNet (ELN) classification, were revised in 2022. The application of the new information in childhood AML will be upcoming in the next few years. The frequency of each genetic abnormality in adult and childhood AML is different; therefore, in this review, we emphasize well-known genetic subtypes in childhood AML, including core-binding factor AML (CBF AML), *KMT2Ar* (*KMT2A*/11q23 rearrangement) AML, normal karyotype AML with somatic mutations, unbalanced cytogenetic abnormalities AML, *NUP98 11p15/NUP09* rearrangement AML, and acute promyelocytic leukemia (APL). Current risk group classification, the management algorithm in childhood AML, and novel treatment modalities such as targeted therapy, immune therapy, and chimeric antigen receptor (CAR) T-cell therapy are reviewed. Finally, the indications of hematopoietic stem cell transplantation (HSCT) in AML are discussed.

## 1. Introduction

Acute leukemia (AL) stands as the most common form of cancer observed among children. Among childhood leukemias, the majority (around 80%) are acute lymphoblastic leukemia (ALL), while a smaller proportion (15–20%) are diagnosed as acute myeloid leukemia (AML) [1]. AML originates from cancerous stem cell precursors, which typically mature into myeloid cells including white blood cells, red blood cells, and platelets. This results in the excessive production of cancerous myeloid stem cells. While AML is relatively rare, occurring at a frequency of only seven cases per million children annually [2,3], its prognosis tends to be poorer compared to other childhood cancers, often marked by a high rate of relapse [4].

Progress in understanding AML at the genetic and molecular levels has led to a better grasp of patient risk and categorization. The WHO classification of myeloid neoplasms and the International Consensus Classification (ICC) of Myeloid Neoplasms and Acute Leukemias were both revised in 2022 [5,6]. Also, the risk group classification, the European LeukemiaNet (ELN) classification, was updated in 2022 [7]. As a result, novel treatment approaches have experienced a rapid advance, leading to improved results in the management of childhood AML over the last thirty years. Currently, complete remission rate has exceeded 90%, and 10-year overall survival rate has also surpassed 60% [8]. The Taiwan Pediatric Oncology group has analyzed the percentages of different subtypes in childhood AML and their survivals as summarized in Figure 1 and Figure 2, respectively [9]. The challenge with AML lies in its susceptibility to a variety of mutations and epigenetic changes that hinder treatment response and elevate the chances of relapse [10].

Despite the integration of findings from adult AML studies into the pediatric context, disparities between childhood and adult AML still persist. Variations in how treatment regimens are tolerated underscore the need for meticulous consideration and tailoring of treatment strategies for childhood AML [4,11]. In this article, we explore several widely acknowledged subtypes of pediatric AML and address important prognostic indicators. Furthermore, both current and prospective treatment strategies are also discussed in detail.

## 2. Well Known Subtypes of Childhood AML

### 2.1. Core-Binding Factor AML

The largest subgroup within pediatric AML is core-binding factor AML (CBF AML), comprising around 25% of cases, typically affecting individuals with a median age of 8–9 years [2]. The prominent mutations in CBF AML include t(8;21)(q22;q22) and inv(16)(p13q22), also recognized as t(16:16)(p13;q22), often denoted as t(8;21) and inv(16) [12]. These genetic anomalies give rise to fusion genes *RUNX1-RUNX1T1* and *CBFβ-MYH11*, respectively, which disrupt the CBF complex (consisting of RUNX1-3 + CBFβ), crucial for hematopoiesis. These fusion genes serve as indicators to assess minimal residual disease (MRD) [13]. Notably, differences exist in clinical presentation, morphology, and cytogenetics between t(8;21) and inv(16) AML [14]. While t(8;21) commonly exhibits a slight increase in leukocyte count and reduced levels of bone marrow primitive cells, leading to thrombocytopenia and anemia, inv(16) tends to affect areas such as the skin, lungs, or central nervous system, manifesting as hepatosplenomegaly, lymphadenopathy, and gingival hyperplasia [14]. However, the CBF fusion gene is essential but often insufficient to trigger leukemogenesis [15]. The second-hit hypothesis proposes that secondary mutations in *RAS-KIT* and *FLT3* are commonly observed molecular alterations in CBF AML [12]. The therapeutic approach for CBF AML includes a regimen of high-dose cytarabine, fludarabine, and gemtuzumab ozogamicin (GO), yielding estimated 10-year survival rates ≥ 75% and notably improving prognosis [16,17]. Overall survival and remission rates are greatly enhanced when the FLAG-GO regimen (fludarabine, cytarabine, granulocyte colony-stimulating factor with GO) is employed according to a clinical trial [18]. Additionally, AML cases with *KIT* mutations experienced bettered disease survival rates through the incorporation of dasatinib [19]. Combining venetoclax with hypomethylating agents has proven effective against inv(16) cases [13]. Individuals diagnosed with t(8;21)-AML have demonstrated favorable results through treatment regimens that encompass increased doses of etoposide, anthracyclines, and cytarabine during the induction phase. Additionally, cumulative elevated doses of cytarabine and etoposide have also contributed to positive treatment outcomes [20].

### 2.2. KMT2Ar (KMT2A/11q23 Rearrangement)

*KMT2A*, situated on chromosome 11q23, is a pivotal player in normal hematopoietic development. Mutations within *KMT2A* constitute 16–20% of pediatric AML cases. Applying the FAB classification, approximately 73% of *KMT2A*rAML falls under the categories of AML M4 or M5. Importantly, *KMT2A*r AML is more common in infants, with a prevalence of 47–55% observed in children aged below 2 years [21]. The prognosis for infant AML featuring *KMT2A*r mutations is generally unfavorable [22]. The first subtype of *KMT2A*r AML enrolled in the 2016 WHO classification is AML characterized by t(9;11)(p21.3;q23.3) *KMT2A-MLLT3* [23]. In the 2022 WHO classification, AML with KMT2A rearrangement replaces “AML with t(9;11)(p22;q23); *KMT2A-MLLT3*”, because over 80 *KMT2A* fusion partners have been described, with *MLLT3*, *AFDN*, *ELL*, and *MLLT10* being most common. The fusion partner identified could provide prognostic information, which may have an impact on monitoring disease [24]. The t(11;19)(q23;p13) translocation, which includes *ELL* (19p13.1) or *MLLT1* (*ENL*) (19p13.3) partners, contributes to 8% and 6% of *KMT2A*r cases, respectively, and carries intermediate prognosis [25]. Another subtype, t(1;11)(q21;q23)/*KMT2A-MLLT11*(AF1Q), shows promise with a favorable prognosis, yet its rarity prevents conclusive assessment. On the other hand, gene fusions associated with unfavorable prognoses encompass cases with t(10;11)(p11.2;q23)/*KMT2A-ABI1*, t(10;11)(p12;q23)/*KMT2A-AF10*, and t(6;11)(q27;q23)/*KMT2A-MLLT4*, constituting 2%, 16%, and 6% of KMT2Ar instances, respectively [8]. The prognostic significance of most *KMT2A* fusions is not certain in childhood AML, but t(4;11)(q21;q23.3)/*KMT2A-MLLT2*, t(6;11)(q27;q23)/*KMT2A-MLLT4*, t(10;11)(p12;q23)/*KMT2A-AF10*, and t(10;11)(p11.2;q23)/*KMT2A-ABI1* were identified with a poor prognosis in most of the literature [26]. Genetic elements *MLLT3*, *ELL*, *MLLT10* (10p12), and *ABI1* (10p11.2) are often linked to infant AML instances. In contrast, *MLLT4* (6q27) is more commonly found in older children, with a median age of 12 years [13,22,27]. Regarding treatment, augmenting conventional chemotherapy with gemtuzumab ozogamicin (GO) and combining it with hematopoietic stem cell transplantation (HSCT) has exhibited enhanced outcomes for *KMT2A*-r AML [28]. Another treatment scheme involves venetoclax, a B-cell lymphoma-2 (BCL-2) blocker, which reactivates the caspase-dependent apoptosis in AML. Recent studies have pointed out that the concurrent use of venetoclax with I-BET151, sunitinib, or thioridazine significantly decreases cell viability in cases of *KMT2A*r AML [29].

### 2.3. Normal Karyotypes

Approximately 22–26% of childhood AML cases exhibit normal karyotypes. These patients are categorized into an intermediate-risk group, displaying varying prognoses. Notably, the proportion of AML cases with a normal cytogenetic profile in children (15–30%) is smaller compared to adults (40–47%) [30]. Risk assessment is determined by established prognostically significant mutations and cytogenetic abnormalities like *NPM1*, *FLT3*-ITD, and *CEBPA*dm. It is important to recognize that a significant portion of AML cases categorized as having a normal karyotype through conventional cytogenetic analysis may obscure other molecular mutations and rearrangements of diagnostic importance, including *FLT3* internal tandem duplications (*FLT3*-ITD), mixed-lineage leukemia gene-partial tandem duplications (*MLL*PTD), or nucleophosmin (NPM1) mutations [31,32]. Among these, *FLT3*-ITD mutations, present in 20–25% of childhood AML cases with an otherwise regular karyotype, stand as a crucial marker for poor prognosis [31,33]. *MLL*-PTD is another mutation linked to poorer prognosis, leading to lowered overall survival (OS) and event-free survival (EFS) in pediatric AML [13]. Conversely, NPM1 mutations occur in 20–30% of normal karyotypepediatric cases and typically result in a positive outlook [32,34]. Notably, the coexistence of *NPM1* mutations with *FLT3*-ITD is correlated with a positive prognosis, whereas *WT1* mutations alongside *FLT3*-ITD exacerbate the prognosis [13].

### 2.4. Unbalanced Cytogenetic Abnormalities in Childhood AML

About 40% of childhood AML cases involve unbalanced cytogenetic abnormalities, with notable prognostic implications like monosomy 5, del(5q), and monosomy 7. However, despite their poor prognostic influence, these abnormalities are detected in just 5% of cases [13]. Monosomy 7 and del(7q), prevalent in childhood myelodysplastic syndromes (MDS), encompass 40% of cases in that context. Notably, del(7q) is more commonly related to CBF leukemia, while monosomy 7 is frequently linked to inv(3)(q21q26). The unfavorable prognosis tied to monosomy 7 mainly arises from its heightened resistance to induction therapy, yielding complete remission rates of 71% to 83% [21,35,36]. Meanwhile, 10–15% of MDS patients and 40% of secondary AML patients experience del(5q). Lenalidomide treatment effectively targets CSNK1A1 for ubiquitin-mediated degradation in isolated del(5q) MDS cases [37]. Unbalanced cytogenetic abnormalities, like del(5q), have emerged as secondary mutations in two AML subtypes with pre-existing cryptic abnormalities: t(5;11)(q35;p15)/*NUP98-NSD1* and the infrequent t(7;21)(p22;q22)/RUNX1-USP42. This underscores the significance of comprehensive cytogenetic and molecular assessments for such cases [13]. Trisomy 8, affecting roughly 10–14% of pediatric AML cases, can appear as the only structural alteration or alongside other genetic irregularities. This anomaly is more frequent in older children (with a median age of 10.1 years) and is often associated with *FLT3*-ITD mutations [38].

### 2.5. NUP98 11p15/NUP09 Rearrangement

Nucleoporin 11p15/98Kd (NUP98) rearrangements are infrequent, occurring in approximately 3–5% of childhood AML cases and occasionally in young adults. Several different partners for NUP98 have been identified, with the most prevalent being the nuclear receptor-binding SET domain protein 1 (NSD1) gene (5q35), present in around 75% of pediatric NUP98r patients. In terms of molecular aspects, CDK6 emerges as a prominently expressed target directly regulated by NUP98-fusion proteins. Intervening with CDK6 activity leads to cell cycle arrest, myeloid differentiation, and apoptosis both in vitro and in vivo. This prompts the exploration of CDK6 suppression as a potential tactic to address NUP98 fusion-associated AML [39]. Additionally, the inhibition of Menin-MLL1 (for instance, VTP50469) stands as another potential therapeutic approach. Studies have demonstrated that such inhibitors promote the upregulation of differentiation markers like CD11B and the downregulation of proleukemogenic transcription factors in NUP98-HOXA9 and NUP98-JARID1A mouse leukemic cell lines, indicating their promise as treatment options [40].

### 2.6. Acute Promyelocytic Leukemia (APL)

Acute promyelocytic leukemia (APL) is an extensively studied variant of AML and is identified by the translocation t(15;17)(q24;q21), which leads to the generation of the *PML-RARA* rearrangement [41]. APL is a relatively uncommon ailment, responsible for 5%–10% of AML cases. Across all age groups, the United States witnesses fewer than 1000 instances of APL annually [42]. The mutated PML-RARA fusion gene prompts the uncontrolled proliferation and accumulation of leukemic white blood cells (WBCs) that remain arrested at the promyelocyte stage, failing to mature or differentiate. The proliferation of these abnormal cells eventually displaces normal blood cell precursors within the bone marrow, resulting in common symptoms like anemia, prolonged bleeding, and recurrent infections. A serious complication of APL is disseminated intravascular coagulation (DIC) [43]. While around 90% of APL cases exhibit the characteristic t(15;17) *PML-RARA* reciprocal translocation, there are instances of other translocations involving the *RARA* gene and other genes apart from *PML*. The presence of *PML-RARA* can be determined through techniques like RQ-PCR (real-time quantitative polymerase chain reaction), FISH (fluorescence in situ hybridization), and chromosome analysis [44].

## 3. Current Treatment Regimens and Future Treatments

Present treatment protocols for AML involve utilizing high doses of cytarabine and anthracycline (cytarabine infused continuously for 7 days with three once-daily injections of an anthracycline, 7 + 3 regimen) to achieve complete remission. However, this approach comes with significant drawbacks such as a heightened risk of infection and cardiac dysfunction. Furthermore, primary chemotherapy faces substantial resistance across various AML subtypes, including *DNMT3A*, *TET2*, *IDH1/2* (more prevalent in adults), and *UP98, WT1, RUNX1, MLLT10, SPECC1,* and *KMT2A* [45]. AML cases with unfavorable genetic factors, as outlined in the European LeukemiaNet (ELN) classification, consistently demonstrate resistance to standard chemotherapy, which consequently leads to an unfavorable prognosis [46]. Rubnitz J. had proposed a risk classification of childhood AML based on important genetic factors (Table 1) and MRD status in 2017, which practically guide the treatment strategies for pediatric AML management (Figure 3) [8]. The genetic abnormalities with prognostic significance in childhood AML were revised in 2022, in which “the intermediate or unknown”, including t(9;11)(p12;q23)/*MLLT3-KMT2A* and t(1;22)(p13;q13)/*RBM15-MKL1*, was eliminated from the list (Table 1). *NPM1* mutation with or without *FLT3*-ITD and *CEBPA* mutation with or without *FLT3*-ITD were categorized as the favorable group and *FLT3*-ITD without *CEPBA* or *NPM1* mutation was categorized as the unfavorable group in this version [26]. The 2022 ELN risk classification by genetics at initial diagnosis categorized mutated *NPM1* with *FLT3*-ITD as the intermediate group [7]. The FLT3-ITD allelic ratio is no longer taken into consideration in the risk classification due to difficulties in standardizing the assay of measurement for the FLT3-ITD allelic ratio, the changing impact of midostaurin-based therapy on FLT3-ITD without NPM1 mutation, and the increasing role of minimal residual disease monitoring in treatment guidance. The suggested treatment algorithm for childhood AML was also revised and enrolled a new entity of relapse/refractory AML in 2021 (Figure 4) [26].

### 3.1. Target Therapies

#### 3.1.1. Gemtuzumab Ozogamicin

Gemtuzumab ozogamicin (GO) is a CD33 targeting monoclonal antibody attached to a toxin. Studies indicate that using GO in induction therapy can reduce relapse risk, though upfront use was linked to increased treatment-related mortality [20]. Although not currently considered for treatment decisions [47], genetic factors such as elevated CD33 expression, *KMT2A* rearrangements, *FLT3*-ITD abnormalities, and single-nucleotide polymorphisms in *ABCB1* could benefit from GO treatment [48,49].

#### 3.1.2. CD123 Target Therapy

CD123 (the IL-3 receptor α-chain), present in many AML cases including leukemic stem cells, is a promising target for therapy. While still in early phases, CD123-targeted treatments offer a potential avenue for treatment [4].

#### 3.1.3. FLT3 Inhibitors

FLT3 inhibitors are a class of molecules that attach to the ATP-binding site on FLT3, resulting in competitive inhibition of kinase activity [50]. These inhibitors come in two classes, type I and type II, both of which target internal tandem duplication (ITD) mutations in *FLT3*. Type I inhibitors, in addition to being active against tyrosine kinase domain (TKD) mutations, possess a higher binding affinity for FLT3 [51]. While at least eight FLT3 inhibitors are under development, none have yet received approval for pediatric use [4]. Sorafenib was among the first multikinase type II inhibitors to be administered for AML treatment. Studies indicated that when used in synergy with chemotherapy, sorafenib yielded better complete remission (CR) and event-free survival (EFS) results in younger adult AML patients, regardless of *FLT3* mutation status. Nonetheless, this combination also raised concerns regarding increased toxicity [52,53]. Sorafenib was proven to be effective and tolerable for children with relapsed AML when administered along chemotherapy in early-phase trials. As a result, sorafenib could be safely integrated into conventional AML chemotherapy, potentially enhancing outcomes for pediatric high allelic ratio (HAR) *FLT3*/ITD+ AML cases [54]. Moreover, sorafenib maintenance therapy was found to reduce relapse risk and mortality following hematopoietic cell transplantation (HCT) for FLT3-ITD-positive AML [55]. Midostaurin, a first-generation type I inhibitor that also influences KIT, demonstrated a capacity to decrease blast percentages on its own in patients with relapsed/refractory AML, irrespective of *FLT3* mutation status [56]. The extensive international RATIFY study illustrated that combining midostaurin with standard induction therapy followed by consolidation using high-dose cytarabine led to longer overall survival (OS) and event-free intervals compared to chemotherapy alone in newly diagnosed AML patients with *FLT3* mutations [46,57].

Gilteritinib, a second-generation type I FLT3 inhibitor, exerts inhibition on AXL, a tyrosine kinase that augments FLT3 activation and contributes to FLT3 inhibitor resistance. Although research involving pediatric patients is currently limited, forthcoming trials are going to explore the application of gilteritinib combined with chemotherapy in FLT3-mutated AML, in both relapse and upfront settings [50]. Notably, highly effective FLT3-specific inhibitors such as quizartinib, gilteritinib, and midostaurin are now accessible for clinical application. In the randomized phase 3 RATIFY trial, integrating typical induction therapy with midostaurin led to extended OS and event-free intervals compared to administering only chemotherapy in individuals newly diagnosed with *FLT3*-mutated AML [57].

#### 3.1.4. BCL-2 Inhibitors

BCL-2 inhibitors are significant in AML treatment. One such inhibitor, venetoclax, targets the antiapoptotic protein BCL-2 abundant in AML cells. There exists evidence indicating that the amalgamation of venetoclax and azacytidine might offer a secure and encouraging treatment alternative for pediatric patients [58]. In a phase 1 study involving continuous 28-day cycles, participants were administered venetoclax orally once a day in combination with IV cytarabine. The study’s analysis suggests that the combination of venetoclax and chemotherapy exhibited both safety and effectiveness in heavily relapsed or refractory pediatric AML. This implies the need for further exploration of this combination in newly diagnosed high-risk pediatric AML patients [59]. Notably, the study also reported a remarkable complete response (CR) rate of 70% when venetoclax was administered alongside cytarabine, regardless of the use of idarubicin. Preclinical studies further indicate that the combination of venetoclax with FLT3 inhibitors such as midostaurin and gilteritinib could have a synergistic effect in triggering apoptosis in AML cells, potentially opening up new possibilities for therapy [60].

#### 3.1.5. AKT/MAPK/STAT Inhibitors

The phosphatidylinositol 3-kinase (PI3K)/Akt/mammalian target of rapamycin (mTOR) signaling pathway assumes a central role in regulating cell proliferation, growth, and survival under normal physiological circumstances. This pathway, along with other significant signals like RAS-MAPK-ERK and JAK2-STAT5, serves as a downstream response to various tyrosine kinases, such as FLT3 and ABL1. Amplification of the PI3K-AKT-mTOR pathway is evident in a notable proportion, roughly around 60%, of individuals with AML [61]. In the realm of AML treatment, three JAK inhibitors—namely pacritinib, ruxolitinib, and lestaurtinib—have been subject to clinical evaluations for both AML and high-risk myeloproliferative neoplasms (MPNs) [46]. Pacritinib, administered to a small group of patients dealing with relapsed/refractory AML, demonstrated a clinical benefit rate of 43% [62].

#### 3.1.6. Menin-KMT2a Inhibitors

AML characterized by *KMT2A* rearrangements (*KMT2A*r) exhibits heightened expressions of HOXA9 and MEIS1 genes. These elevated levels arise from the interaction between oncogenic KMT2A fusion proteins and complex-forming counterparts like LEDGF, DOT1L, and menin. Recent efforts have focused on strategies to disrupt this link between KMT2A protein and menin [4]. Notably, the initial phase 1 study of the menin-KMT2A inhibitor SNDX-5613 (Revumenib) as a standalone treatment showcased promising outcomes among patients with relapsed/refractory AML who carried *KMT2A* rearrangements or *NPM1* mutations [63].

#### 3.1.7. Anti-CD47 Antibody

CD47, a cell surface protein that is widely distributed, has a pivotal role in regulating phagocytosis via innate immune cells like macrophages and dendritic cells. This modulation takes place via the engagement of CD47 with SIRP-alpha, a receptor found on these immune cells. This interaction imparts a suppressive signal that hinders phagocytosis [64]. As such, prognosis may be unfavorable when AML stem cells express CD47 [65]. In terms of treatment, the anti-CD47 antibody magrolimab, when administered along azacitidine, demonstrated a remarkable 75% overall response rate in adults experiencing untreated higher-risk myelodysplastic syndromes (MDS) [66]. An important step in research is the ongoing phase 2 study led by Gilead, which investigates the therapeutic possibilities of magrolimab when combined with different cytotoxic substances for the treatment of myeloid malignancies.

### 3.2. Chimeric Antigen Receptor (CAR) T-Cell Therapy

CAR-T cell therapy faces challenges in AML due to the absence of a unique cancer surface marker. However, two cell markers, CD33 and CD123, are especially common on AML cells, with CD33 detected in 90% of instances, and CD123 found in 75% of AML cases. Notably, only a minimal proportion (<5%) lacks both [67]. While CD33 therapies can affect non-cancerous myeloid cells, the absence of CD33 on hematopoietic stem cells, combined with the favorable results seen with GO, provides a rationale for investigating CD33-targeting CAR-Ts. CD123, found excessively in various hematologic malignancies, serves as an attractive target across leukemia types. Yet, its presence on healthy hematopoietic stem cells suggests it might be best utilized as an intermediary step toward SCT. Ongoing studies involving CAR-T therapies that target CD33 and CD123 are being conducted in pediatric cases of relapsed or refractory AML [68,69,70,71]. It is important to note that immunotarget expression in pediatric AML deviates from adult AML. In pediatric AML, CLEC12A and CD33 stand out as preferred generic combined immunotargets, while CD33 and FLT3 serve as immunotargets specific to *KMT2A*-mutated infant AML [72].

### 3.3. Checkpoint Inhibitors

Checkpoint inhibitors such as pembrolizumab and nivolumab, which target PD-1, along with ipilimumab targeting CTLA-4, emerge as promising therapy options. Clinical trials are investigating the synergism between the above mentioned PD-1 inhibitors and anti-PD-L1 antibodies including atezolizumab and durvalumab [73,74].

### 3.4. Epigenic Therapies

#### 3.4.1. Hypomethylating Agents (HMAs)

Epigenetic therapies encompass hypomethylating agents (HMAs) like decitabine and azacitidine, which inhibit DNA methyltransferases, inducing tumor suppressor expression and apoptosis [75,76]. St. Jude AML16 trial (NCT03164057) evaluates HMAs alongside cytotoxic chemotherapy [77,78].

#### 3.4.2. Histone Deacetylase (HDAC) Inhibitors

Histone deacetylase (HDAC) inhibitors like vorinostat, panobinostat, tricostatin regulate chromatin by removing acetyl groups from histones, impacting gene transcription, and promoting apoptosis. Synergy with standard chemotherapy is evident [79]. Panobinostat combined with hypomethylating agents and chemotherapy is promising [80]. HDAC inhibitors combine safely with agents like DNMT inhibitors in adult studies and show potential in relapsed pediatric AML [81,82].

### 3.5. TP53 Stabilizers

The TP53 protein plays a crucial role as a tumor suppressor by regulating cell division and apoptosis. TP53 mutations exist in around 30–40% of secondary AML cases and less than 10% of normal AML cases [39]. There are small molecules, such as PRIMA-1, that reactivate mutant TP53. PRIMA-1 and its methylated form induce apoptosis in *TP53*-mutated cancer cells by restoring wild-type conformation and specific DNA binding of mutant p53 [83,84]. A clinical investigation that combined APR-246 and azacitidine resulted in a 71% overall response rate among patients diagnosed with TP53-mutated high-risk MDS or AML [85]. APR-548, an upcoming TP-53 stabilizer, is under examination in a phase 1 clinical trial (NCT04638309) by Aprea Therapeutics.

### 3.6. Repurposing of Chemotherapy Drugs

#### 3.6.1. CPX-351

CPX-351 involves a liposomal encapsulation of cytarabine and daunorubicin, combined in a synergistic 5:1 drug ratio. In older patients aged 60 to 75 years with newly diagnosed secondary AML, CPX-351 treatment is associated with significantly longer survival compared to conventional 7 + 3 regimen [86]. In pediatric cases of relapsed or refractory AML, a phase I/II trial revealed significant results, as 75% achieved complete remission (CR) following a single CPX-351 treatment cycle. Among these, 80% showed no signs of minimal residual disease (MRD) [87].

#### 3.6.2. Oxidative Phosphorylation Inhibitors: Atovaquone Repurpose

Atovaquone, an anti-infective agent, is typically administered to prevent and to treat pneumonia, particularly pneumocystis jiroveci pneumonia (PJP), a common concern in AML affected children. Atovaquone also hampers oxidative phosphorylation and thereby helps combat AML in patient-derived xenograft mouse models [88,89].

### 3.7. Treatment for Acute Promyelocytic Leukemia (APL)

ATRA (all-trans retinoic acid) combined with ATO (arsenic trioxide) and induction chemotherapy represents the current treatment approach for APL, resulting in significantly improved outcomes compared to previous years [41]. This combination has led to high cure rates, with an OS of around 95% and EFS of 90% for pediatric APL patients. The treatment induces the differentiation of APL blasts [90]. Prognosis and treatment response in APL are linked to different PML-RARA isoforms, but the evidence from studies is inconsistent, so current guidelines recommend not altering standard therapy based on isoforms [91]. Recent progress in APL therapy has shifted focus towards minimizing unnecessary treatments, improving treatment tolerance, and enhancing quality of life [42].

ATRA and ATO additions to initial APL therapy have caused relapsed patients to become insensitive to such treatment. For relapse treatment, options include gemtuzumab ozogamicin (GO), a monoclonal antibody–drug conjugate targeting CD33-expressing cells, which has demonstrated effectiveness in achieving molecular remissions in both newly diagnosed and relapsed patients [92]. Another approach is to investigate a synthetic retinoid called tamibarotene, which boasts an elevated binding affinity for PML-RARα. While tamibarotene has shown promise in reducing relapses in high-risk adult APL patients [93], its application to pediatric APL is still under study. Notably, gemtuzumab ozogamicin is approved for relapsed APL cases in children aged 2 years and older who express CD33.

## 4. Hematopoietic Stem Cell Transplantation (HSCT) Guidelines for AML

### 4.1. Hematopoietic Stem Cell Transplant (HSCT) Recommendations

HSCT guidelines emphasize recipient characteristics, donor selection, and preparatory regimen. Such transplantation aims to decrease relapse rates, consolidate graft-versus-leukemia (GVL) effects, and minimize graft-versus-host disease. If unfavorable cytogenetic properties are observed or response to induction therapy is limited, HSCT will be commonly pursued in the first complete remission (CR1) to intensify treatment due to previously suboptimal results with chemotherapy alone [94,95].

### 4.2. Unfavorable-Risk Molecular Abnormality

In situations involving *FLT3*/ITD, it is recommended that patients proceed to HSCT in CR1, as studies suggest this approach improves outcomes. Co-existing favorable risk mutations, such as *NPM1*, have been found to mitigate the poor prognostic impact of *FLT3*/ITD [96,97]. This phenomenon has been observed in adult studies and preliminary research in pediatric cases, possibly suggesting the potential benefits of HSCT in CR1 for this subset of *FLT3*/ITD patients [98].

### 4.3. Unfavorable-Risk Cytogenetic Abnormality

For cases involving high-risk KMT2Ar, t(6;9)(p23;q34) chromosomal rearrangement, monosomies of 7 and 5 (or del 5q), abnormalities of 3q and 12q, complex karyotype, NUP98-NSD1 fusion, t(5;11)(q35;p15.5), and cryptic chromosome 16 inv(16)(p13.3q24.3), the panel advises HSCT in CR1 for patients with minimal residual disease (MRD) at the end of induction (EOI) stage. If primary induction failure (PIF) persists beyond the second EOI, HSCT should be considered after a third chemotherapy cycle, as achieving complete remission (CR) with further chemotherapy becomes unlikely and risks of toxicity increase. However, factors like emerging therapeutic strategies and disease progression should also guide HSCT timing decisions. Children with therapy-related AML (tAML) are prone to relapse and respond poorly to multiple cycles of chemotherapy due to prior exposure. Early HSCT consideration, generally following 1 or 2 chemotherapy cycles to elicit remission, is advised for these cases [99].

## 5. Conclusions

The treatment of pediatric AML still poses many challenges. Our understanding of the genetics and molecular characterization of AML has led to a greater understanding of this disease with better patient risk and categorization. Novel treatment approaches with new therapeutics are also in clinical trials and show promising results. We hope that these new therapies can soon lead to less toxic, more effective and targeted treatment therapies that can help pediatric patients especially with resistant or refractory disease.

## Figures and Tables

**Figure 1 pharmaceuticals-16-01614-f001:**
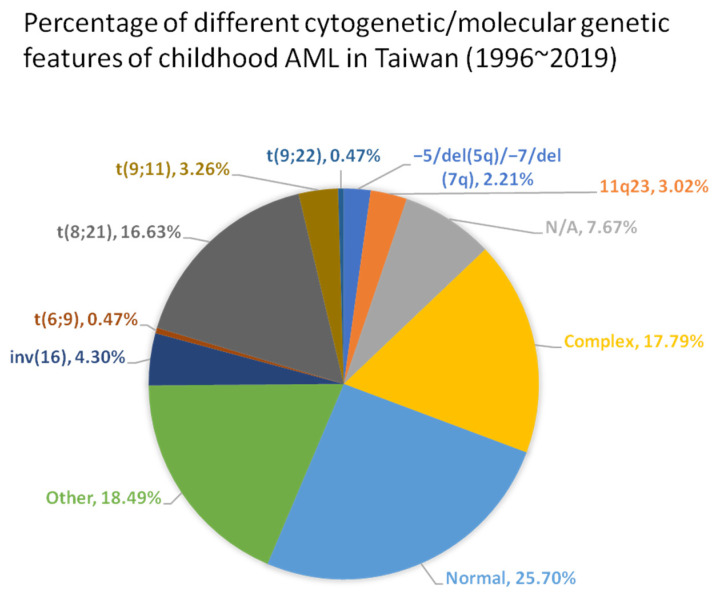
Percentage of different cytogenetic/molecular genetic features of childhood AML in Taiwan (1996~2019). The pie chart showed the percentages of different genetic subtypes in childhood AML diagnosed between 1996 and 2019. The total number of patients enrolled was 860. (Adapted from Table 2 in Yang et al., 2021 [9]).

**Figure 2 pharmaceuticals-16-01614-f002:**
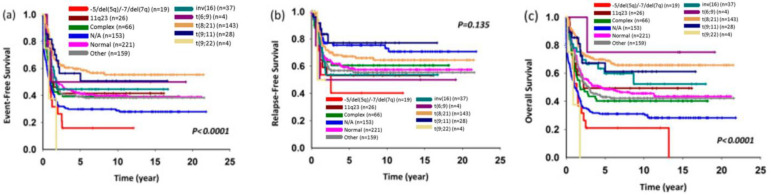
Five-year (**a**) event-free survival, (**b**) relapse-free survival, and (**c**) overall survival rates according to major cytogenetic alterations. (From Figure 2 in Yang et al., 2021 with permission [9]).

**Figure 3 pharmaceuticals-16-01614-f003:**
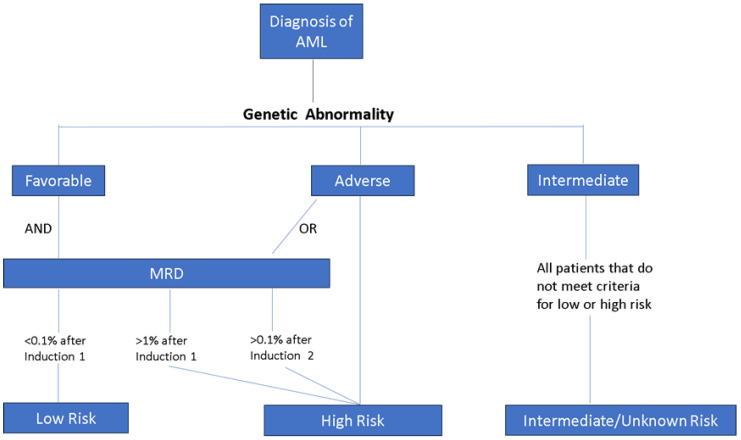
Recommended risk classification based on genetic alterations using prognostically important genetic abnormalities in childhood AML and MRD (minimal residual disease), adapted from [8].

**Figure 4 pharmaceuticals-16-01614-f004:**
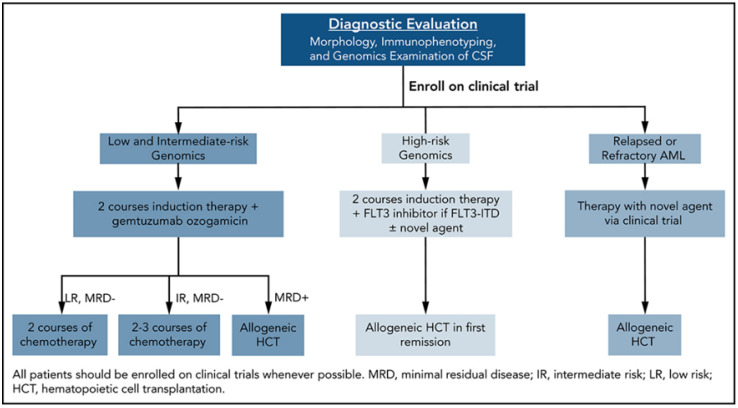
Recommended treatment of childhood AML [26]. Reprinted from Blood, Vol 138 Issue 12 Pages 1009–1018, Jeffrey E. Rubnitz, Gertjan J.L. Kaspers, How I treat pediatric acute myeloid leukemia, Visual abstract, 2021, with permission.

**Table 1 pharmaceuticals-16-01614-t001:** Prognostically important genetic abnormalities in pediatric acute myeloid leukemia proposed in 2017 and 2021 [8,26].

	2017	2021
Favorable	t(8;21)(q22;q22)/*RUNX1-RUNX1T1*	t(8;21)(q22;q22)/*RUNX1-RUNX1T1*
	inv(16)(p13.1;q22)/*CBFb-MYH11*	inv(16)(p13.1;q22)/t(16;16)(p13.1;q22)/*CBFB-MYH11*
	t(16;16)(p13.1;q22)/*CBFb-MYH11*	
	Mutated *NPM1* without *FLT3*-ITD	*NPM1* mutation with or without *FLT3*-ITD
	Biallelic mutations of *CEBPA*	*CEBPA* mutation with or without *FLT3*-ITD
	t(1;11)(q21;q23)/*MLLT11-KMT2A*	
Unfavorable	t(6;11)(q27;q23)/*MLLT4-KMT2A*	inv(16)(p13.3q24.3)/*CBFA2T3-GLIS2*
	t(10;11)(p12;q23)/*MLLT10-KMT2A*	t(10;11)(p12;q23)/*KMT2A-AF10*
	t(10;11)(p11.2;q23)/*ABI1-KMT2A*	t(10;11)(p11.2;q23)/*KMT2A-ABI1*
	t(6;9)(p23;q34)/*DEK-NUP214*	t(6;11)(q27;q23)/*KMT2A-MLLT4*
	t(8;16)(p11;p13)/*KAT6A-CREBBP*	t(4;11)(q21;q23.3)/*KMT2A-MLLT2*
	t(16;21)(q24;q22)/*RUNX1-CBFA2T3*	t(11;12)(p15;p13)/*NUP98-KDM5A*
	t(5;11)(q35;p15.5)/*NUP98-NSD1*	t(7;11)(p15.4;p15)/*NUP98-HOXA9*
	inv(16)(p13.3q24.3)/*CBFA2T3-GLIS2*	t(5;11)(q35;p15)/*NUP98-NSD1*
	t(11;15)(p15;q35)/*NUP98-KDM5A*	t(6;9)(p23;q34)/*DEK-NUP214*
	t(3;5)(q25;q34)/*NPM1-MLF1*	t(8;16)(p11;p13)/*KAT6A-CREBBP*
	*FLT3-ITD*	t(16;21)(q24;q22)/*RUNX1-CBFA2T3*
	Monosomy 7	t(7;12)(q36;p13)/*ETV6-HLXB*
		t(3;21)(26.2;q22)/*RUNX1-MECOM*
		t(16;21)(p11.2;q22.2)/*FUS-ERG*
		*FLT3*-ITD without *CEPBA* or *NPM1* mutation
		inv(3)(q21.3q26.2)/t(3;3)(q21.3q26.2)/*RPN1-MECOM*
		t(3;5)(q25;q34)/*NPM1-MLF1*
		t(10;11)(p12.3;q14.2)/*PICALM-MLLT10*
		−7, −5, 5q-
Intermediate or unknown	t(9;11)(p12;q23)/*MLLT3-KMT2A*	
	Other *KMT2A* fusions	
	t(1;22)(p13;q13)/*RBM15-MKL1*	

## Data Availability

Data sharing is not applicable.

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
