# Peer review of "A Review of Childhood Acute Myeloid Leukemia: Diagnosis and Novel Treatment"

_pharmaceuticals, 2023, doi:10.3390/ph16111614_

Round 1
Reviewer 1 Report
Comments and Suggestions for Authors
Tseng and colleagues review the molecular landscape of pediatric AML, focusing on the main the drivers in children. The also describe novel therapies including CAR-T cells and HSCT indication. Overall, the manuscript is will written and organized.
Minor comments:
- The authors do not mention the novel international consensus classification.
- The authors could discuss the implications of FLT3 ratio removal in AML risk stratification.
Comments on the Quality of English LanguageMinor revision needed
Author Response
Dear Reviewer,
We have briefly mentioned the ICC guideline of 2022 and also the fact that FLT3-ITD allelic ratio is no longer taken into consideration in the risk classification of AML.
Reviewer 2 Report
Comments and Suggestions for Authors
In this review manuscript, Tseng et al. summarized current diagnosis, classification, and treatment approaches in childhood acute myeloid leukemia (AML). Although, amongst childhood acute leukemia, AML is not as common as acute lymphoblastic leukemia (ALL), the prognosis of AML is much poorer than that of ALL. This underscores the unmet medical need to improve childhood AML therapy. Therefore, such review paper will provide the current status of childhood AML diagnosis and therapy, which may contribute to the fields of fundamental and clinical studies focusing on childhood AML.
The manuscript will benefit from the inclusion of a pie chart showing the percentages of different subtypes in childhood AML. Additionally, it would be much better to summarize the survival rate of childhood AMLs with different genetic backgrounds.
Table 2 summarized the chemotherapy drugs utilized for acute promyelocytic leukemia (APL). With current therapy, APL patients achieve approximately 95% OS. Thus, list of drugs for APL therapy could not demonstrate clinical significance. It would be much more helpful to summarize chemotherapy drugs against the childhood AMLs with poor prognosis, such as KMT2A-rearrangements.
Comments on the Quality of English LanguageThe authors should correct English grammar error and typos throughout the manuscript.
Author Response
Dear Reviewer,
We have added the percentage of different subtypes of childhood AML and their respective survivals, as analyzed by the Taiwan Pediatric Oncology group. We have also taken out the table of chemotherapy drugs for APL as suggested.
Thank you
Reviewer 3 Report
Comments and Suggestions for Authors
The review is written well and shows the progress in treatment of children`s AML. I do not have significant remarks except one in verse 363 the authors write that CPX-351 "was designed to selectively target leukemia cells" and this is not true. Stoichiometry was preserved like in 3+7 regimen and it appeared to have better results when cytostatics were given at once together in liposome.
Author Response
Dear Reviewer,
We have taken out the incorrect information and added the following. CPX-351 involves a liposomal encapsulation of cytarabine and daunorubicin, combined in a synergistic 5:1 drug ratio. In older patients age 60 to 75 years with newly diagnosed secondary AML, CPX-351 treatment is associated with significantly longer survival compared to conventional 7+3 regimen
Thank you
Round 2
Reviewer 2 Report
Comments and Suggestions for Authors
The authors have addressed my questions. I have no more questions.